# Immunostimulatory Effect of Postbiotics Prepared from *Phellinus linteus* Mycelial Submerged Culture via Activation of Spleen and Peyer’s Patch in C3H/HeN Mice

**DOI:** 10.3390/ph15101215

**Published:** 2022-09-30

**Authors:** Min Geun Suh, Hyun Young Shin, Eun-Jin Jeong, Gaeuleh Kim, Se Bin Jeong, Eun Ji Ha, Sang-Yong Choi, Sung-Kwon Moon, Kwang-Won Yu, Hyung-Joo Suh, Hoon Kim

**Affiliations:** 1Department of Integrated Biomedical and Life Sciences, Korea University, 145 Anam-ro, Seongbuk-gu, Seoul 02841, Korea; 2R&D Center Neo Cremar Cooperation Limited, 211 Jungdae-ro, Songpa-gu, Seoul 05702, Korea; 3BK21FOUR R&E Center for Learning Health Systems, Korea University, 145 Anam-ro, Seongbuk-gu, Seoul 02841, Korea; 4Major in Food&Nutrition, Korea National University of Transportation, 61 Daehak-ro, Chungju-si 27909, Korea; 5Department of Food and Nutrition, Chung-Ang University, 17546 Seodong-daero, Daedeok-myeon, Anseong 17546, Korea

**Keywords:** medicinal mushroom, submerged culture-derived postbiotics, intestinal immune system-modulating activity, spleen-mediated immunity, polysaccharide

## Abstract

Medicinal mushrooms are an important natural resource promoting health benefits. Herein, *Phellinus linteus* mycelia were prepared under submerged cultivation, the mycelium-containing culture broth was extracted as a whole to obtain the postbiotic materials (PLME), and its effect on the immune system was evaluated in normal C3H/HeN mice. Oral administration of PLME for 4 weeks was well tolerated and safe. In the PLME-administered groups, in addition to the production of immunostimulatory cytokines, such as interferon gamma (IFN-γ), tumor necrosis factor-α (TNF-α), and interleukin 6 (IL-6), the mitogenic activity was significantly increased. PLME administration also significantly increased the levels of serum immunoglobulin G (IgG) and IgA in the small intestinal fluid and Peyer’s patches and enhanced Peyer’s patch-mediated bone marrow cell proliferation activity and cytokine production (IL-2, IL-6, and IFN-γ). Histomorphometric analyses showed an increase in immune cells in the spleen and small intestinal tissues of mice administered PLME, supporting the rationale for its immune system activation. PLME mainly contained neutral sugar (969.1 mg/g), comprising primarily of glucose as a monosaccharide unit. The β-glucan content was 88.5 mg/g. Data suggest that PLME effectively promote immune function by stimulating the systemic immune system through the spleen and intestinal immune tissues. PLME can thus be developed as a functional ingredient to enhance immune functions.

## 1. Introduction

Mushrooms have been traditionally used as valuable edible and medicinal food ingredients since ancient times [1]. It is well known that traditional Eastern and Western medical systems have long believed in the therapeutic benefits of mushrooms, which, to a large extent, is in line with their long history as a food source [2]. Recently, several medicinal mushrooms have been considered as important natural resources to prevent and/or treat various types of human diseases [3]. Although almost all parts of mushrooms, such as fruiting bodies, sclerotia, mycelia, and spores, have been used for medicinal or health promoting purposes [4,5], in recent years, the submerged culture-based mycelium production and use of the whole culture broth (as a source of postbiotics) have been increasing in various industrial fields including health functional foods, cosmeceuticals, and pharmaceuticals, because of their various advantages, such as low production costs, sustainable and uniform productivity, and eco-friendliness [6,7,8]. Moreover, a considerable number of novel biological metabolites could be produced during mushroom mycelium cultivation process using the submerged culture [9]. However, despite these useful indicators, techniques for producing and utilizing medicinal mycelia through submerged cultivation on a fermenter scale are not widely applied industrially [7].

Recently, interest in postbiotics has increased as a promising alternative to probiotics and prebiotics. In 2021, the International Scientific Association of Probiotics and Prebiotics (ISAPP) proposed a definition of postbiotics as “a preparation of inanimate microorganisms and/or their components that confer a health benefit on the host” [10]. As per this description, postbiotics can refer to microbial cell debris destroyed by extraction, cell fractions, or metabolites, that are naturally produced by living cells during fermentation and that provide various health benefits to the host [11]. Furthermore, the category of postbiotics can include active substances isolated from fruit bodies, mycelia, or culture broth of fungi, including mushrooms. However, only a few studies and applications of postbiotics derived from mushroom mycelia under submerged cultivation have been performed.

*Phellinus linteus*, a member of the Hymenochaetaceae family, has been widely used as a medicinal mushroom for over 2000 years [12]. It is a wood decay fungus that is primarily distributed in Eastern Asian countries and is known as “sangwhang” in Korea, “sanghuang” in China, and “meshimakobu” in Japan [12,13]. Growing evidence indicates that *P. linteus* possesses a wide range of therapeutic effects, including antioxidative, antimicrobial, antiviral, antitumor, anti-inflammatory, antidiabetic, neuroprotective, and immunomodulatory [12,13]. In particular, it has been reported that mycelia of *P. linteus* exhibits a wide range of pharmacological activities, such as hepatoprotective [14,15], anti-inflammatory [16], anti-atopic [17], anti-allergic [18], anti-osteoarthritic [19], and immunomodulatory [16] effects. Nevertheless, pharmacological and therapeutic studies on the whole parts or specific extracts of *P. linteus* mycelium have been explored relatively less than those with fruiting bodies. Considering the environmental and economic benefits of mycelia, more functional and compositional studies of *P. linteus* mycelium or its postbiotics are needed to promote its industrial applicability.

In the present study, we hypothesized that industrially prepared culture broth containing *P. linteus* mycelia might modulate the intestinal immune system. To investigate this, *P. linteus* mycelium was grown under submerged cultivation in an industrial jar fermenter, and the whole culture broth was extracted at a high temperature to obtain a postbiotic from the culture broth. To evaluate the effect of *P. linteus* mycelia-derived postbiotics (PLME) on modulation of the intestinal immune system, we evaluated various physical, biochemical, and immunological changes in C3H/HeN mice following oral administration of PLME. In addition, the general components and sugar composition of PLME were analyzed to estimate the active ingredients responsible for the immunomodulatory activity.

## 2. Results

### 2.1. Effect of PLME Administration on the Major Organs

After PLME were administered to the mice for 4 weeks, three major organs, the heart, kidney, and liver, were harvested from the animals and weighed to evaluate the toxic effect of PLME. As shown in Table 1, no significant differences were observed in the weights of harvested organs among all groups of mice. The organs were sectioned and stained with H & E to further evaluate the histological changes induced by PLME administration. Figure 1 shows representative microscopic images of the stained tissue sections. The results indicated no remarkable histological differences in the heart, kidney, or liver among the groups.

### 2.2. Effect of PLME Administration on the Spleen-Mediated Immunostimulation

The spleen tissue of the mice following final administration of the test constituents was collected, macroscopically observed, and weighed. Figure 2a shows representative images of the whole spleen tissue. The mean weight was 0.066–0.075 g, without a statistically significant difference (data not shown). These results revealed that there were no apparent differences in the appearance and weight of spleen tissue across all groups of mice. The spleen tissue sections were then stained with H & E to assess any morphological differences in the spleen tissue possibly induced by PLME administration (Figure 2b). Control (NOR and CON) and PLME (PLME-L, PLME-M, and PLME-H)-administered mice showed similar intact spleen tissue structures with regular white and red pulps. However, it was noted that the formation of central arteries (yellow arrows) in the white pulp area of spleen tissue was increased in the PLME-administered mice, which implies an increase in periarteriolar lymphoid sheaths that are populated by T cells surrounding the central arteries, suggesting an increase in the lymphocyte population. Next, splenocytes were isolated and cultivated in vitro to evaluate their mitogenic and cytokine production activities. As shown in Figure 3a, the mitogenic activity of animals in the CON and PLME-L groups was not significantly different from that of the NOR group. However, splenocytes isolated from the PLME-M- and PLME-H-administered groups exhibited significantly increased mitogenic activity by 1.8- and 3.0-folds, respectively, compared to those isolated from the NOR group. Figure 3b–d show that various splenocyte-producing cytokines, such as IFN-γ, TNF-α, and IL-6, were produced due to PLME stimulation. Although splenocytes isolated from mice in the PLME-L and PLME-M groups did not show a significant increase in IFN-γ and TNF-α production, their levels increased significantly in splenocytes isolated from PLME-H-administered mice (Figure 3b,c). Interestingly, though IL-6 production was significantly increased in the group of PLME-M-administered splenocytes, there was no significant difference among the other groups when compared with the NOR group.

### 2.3. Effect of PLME Administration on the Immunoglobulin Production

To evaluate the immunoglobulin production by PLME, IgG and IgA levels were measured in the serum and intestinal tissues. Oral administration of PLME significantly augmented serum IgG production (Figure 4a). In particular, serum IgG levels were remarkably increased in the PLME-H group by 2.9-fold as compared to the NOR group. IgA levels were measured in the serum, small intestinal fluid, and culture supernatant of Peyer’s patch cells (Figure 4b–d). As seen from Figure 4b, the serum IgA levels were not significantly different among the groups. In the small intestinal fluid, compared with the NOR group, IgA levels were significantly increased only in the PLME-M group, whereas the levels did not significantly differ in the PLME-L and PLME-H groups (Figure 4c). Mice in the PLME-L group exhibited a significant increase in IgA production in Peyer’s patch cell-culture supernatant; however, the production of IgA was seen to decrease with an increase in the dose administered (Figure 4d).

### 2.4. Effect of PLME Administration on the Intestinal Immune System-Modulating Activity through Peyer’s Patch

To evaluate the effect of PLME administration on the intestinal immune system, bone marrow cell proliferation was measured by inoculating conditioned medium with Peyer’s patch cells isolated from each group of mice. Following 6 days of incubation with the conditioned medium, morphological changes and proliferation activity of bone marrow cells were measured. Representative images of bone marrow cells and the quantified results are depicted in Figure 5a,b, respectively.

Bone marrow cells from the NOR and CON groups had similar morphological characteristics and cell numbers. However, as compared to the NOR group, the bone marrow cells of the PLME-M and PLME-H groups showed differentiated morphologies and significantly increased cell numbers. In particular, the bone marrow cells of the PLME-M group showed the most potent proliferative activity compared to the other groups. To estimate whether PLME administration could induce cytokine production in the intestinal immune system, IL-2, IL-6, and IFN-γ levels were measured in the Peyer’s patch cell culture supernatant (Figure 5c–e). PLME-L administration did not facilitate the measurement of any cytokines in the Peyer’s patch compared with vehicle or medium administration (NOR or CON group). The administration of PLME-M, however, facilitated significant production of only IL-6 and IFN-γ in the Peyer’s patch isolated from mice. Interestingly, in the group administered PLME-H, all cytokines measured in the Peyer’s patch were significantly increased, compared to vehicle or medium administration. Histological changes in the small intestinal tissue were evaluated by H & E staining. As shown in Figure 6, the density of immune cells (red arrows) in the small intestinal tissue of stained sections markedly increased toward the submucosa and muscularis externa sites by PLME administration in a dose-dependent manner. These results suggest that PLME induce immune cell infiltration and activation in intestinal tissue.

### 2.5. General Components and Sugar Composition of PLME

The chemical composition of PLME was determined using phenol-sulfuring acid, m-hydroxyldiphenyl, and Bradford assays (Table 2). PLME was composed of mainly neutral sugars (969.1 mg/g) and minimal amounts of uronic acids (37.1 mg/g) and proteins (2.2 mg/g). To further estimate the sugar composition, PLME were analyzed using HPLC-UVD after derivatization with PMP. The results indicated that PLME contained mainly glucose (93.8%) in addition to small amounts of galactose (2.7%), mannose (1.2%), and arabinose (1.0%) as neutral monosaccharide units, and galacturonic acid (1.0%) as uronic acid. The large amounts of glucose units in PLME led us to determine its β-1,3-1,6-glucan composition. The results showed that PLME contained 88.5 mg/g β-1,3:1,6-glucan.

## 3. Discussion

Although life expectancy continues to increase, concerns about the weakening of immune function arising from aging and maintaining a healthy life are important social concerns in modern society [20]. The recent worldwide pandemic arising from infectious diseases, such as the novel coronavirus disease (COVID-19), has served as an opportunity to awaken to the importance of maintaining the body’s immunity. Human health can be affected by various environmental factors, such as lifestyle, dietary patterns, and hazardous external factors. Specific foods or food-derived constituents can help strengthen the immunity [21]. In addition to their therapeutic efficacy, medicinal mushrooms have been proposed as promising ingredients for safe stimulation of the immunity of the human body [22]. This knowledge, coupled with the recent advancements in cultivation technologies, the mycelial production under submerged cultivation and its utilization as a postbiotic ingredient, has attracted attention as a promising technology that can potentially produce functional and nutritional supplements and serve as immunomodulating agents [9,23].

In the present study, PLME were administered to C3H/HeN mice for four weeks, and their effects on the immune system were evaluated. At all doses administered (250, 500, and 1000 mg/kg bw/ day), PLME-administered mice showed no significant change in body weight compared to the animals in the NOR group (data not shown). Furthermore, the histochemical analysis of major organs, the heart, kidney, and liver, revealed that PLME administration at all the doses tested showed no abnormal changes in weight or histopathology (Table 1 and Figure 1). These results imply that the administration of PLME at the indicated concentrations was well tolerated and safe. Given the lack of in vivo toxicology studies on *P. linteus* mycelia, our study is valuable as it provides the underlying data on its safety aspect.

Next, the effect of the oral administration of PLME on the spleen-mediated adaptive immune system was evaluated. PLME administration did not affect changes in weight and size of spleen tissue compared with the control groups (NOR and CON), but interestingly, the central artery areas of the white pulp appeared to increase after PLME administration (Figure 2). Because the white pulp is considered a key regulating center for adaptive immune activation, our histomorphometric result implicates that PLME administration can stimulate adaptive immune activation without physical alteration of the spleen tissue. Our results also indicate that the oral administration of PLME facilitates the mitogenic activity of splenocytes in a dose-dependent manner (Figure 3), implying that PLME could be applied as a mitogen that triggers lymphocyte mitosis and blastogenesis. Splenocytes are composed of various types of immune cells. Activated splenocytes elicit subsequent immune system activation by releasing various cytokines [24,25]. IFN-γ is a major immunostimulatory cytokine involved in cellular immunity, including both innate and adaptive immune cells [26]. TNF-α plays a crucial role in the development and remodeling of secondary lymphoid tissues, including the spleen [27]. IL-6 is a pleiotropic cytokine that plays multiple roles in the regulation of hematopoiesis, immune responses, and acute phase reactions, and is broadly secreted not only by activated lymphocytes but also by various types of non-lymphoid cells [28]. Many studies have established these cytokines as biomarkers that support immunostimulation through splenocyte activation [25,29]. We confirmed that the production of IFN-γ, TNF-α, and IL-6 was significantly increased in splenocytes by oral administration of PLME (Figure 3b–d), suggesting activation of the spleen-mediated immune system following PLME consumption.

The oral administration of PLME significantly increased the antibody levels (Figure 4). Compared with the NOR group, the levels of serum IgG in the PLME-administered groups were significantly higher, but those of serum IgA were not. The results are as expected because IgG is generally found in the blood circulation while IgA is predominantly found in mucosal tissues, including the intestinal tract. To explore whether PLME administration could activate the intestinal immunity, we further determined the levels of IgA in intestinal tissues, including small intestinal fluid and Peyer’s patch. Unlike the serum IgG level, the levels of IgA in the small intestinal fluid were the highest in the mice administered PLME-M; however, in Peyer’s patch, the levels peaked in animals administered PLME-L. Although the exact reason for antibody production following PLME administration could not be clearly identified in this study, it is possible that PLME administration could increase serum IgG and intestinal IgA levels. Given that a fundamental defense strategy in the intestinal immune system is to produce IgA [30], our results highlight the potential of PLME as an intestinal immunomodulator. In addition to the essential site for producing IgA-producing lymphocytes, the Peyer’s patch is an important site for triggering the antigen-specific immune response against foreign matters [31,32]. Peyer’s patch cells isolated from mice administered with PLME were artificially incubated for five days, and the culture supernatant of Peyer’s patch cells was inoculated into bone marrow cells to evaluate whether the bone marrow cells could differentiate and proliferate through the various factors secreted by Peyer’s patch cells. As expected, we confirmed that bone marrow cells were activated as indicated by morphological changes and an increase in cell numbers (Figure 4a,b), implying that the bone marrow cells were stimulated through Peyer’s patch activation following PLME administration. We also determined cytokine production induced by PLME administration in Peyer’s patches. The results showed that PLME administration dose dependently increased the production of IL-2, IL-6, and IFN-γ, suggesting activation of the intestinal immune system following PLME consumption. Activation of the intestinal immune system was also verified via H & E staining of the intestinal sections, which showed a marked increase in immune cells toward the submucosa and muscularis externa sites (Figure 6). Consequently, our findings suggest that PLME administration activates the immune system mediated by both the spleen and the intestinal immune system.

Finally, the general and sugar compositions of PLME were analyzed to estimate the active constituents (Table 2). It was noted that PLME contain mainly neutral sugar (969.1 mg/g), consisting mainly of glucose units (93.8 mole %) in addition to small amounts of galactose (2.7 mole %) and mannose (1.2 mole %). Because glucans present in fungal and mushroom cell walls are generally composed of short D-glucose side chains branched at the C(O)-6 position of the long β-1,3-glucan backbone [33], we further quantified the β-1,3:1,6-glucan content, rather than β-1,3:1,4-glucan, of PLME. The β-glucan content of PLME was determined to be 88.5 mg/g. It has been generally known that fungal and yeast cells contain more than 50% of β-1,3:1,6-glucans with various structural diversity [33]. Interestingly, our results indicated that PLME have a relatively small composition of β-1,3:1,6-glucans (8.8%), despite the high ratio of glucose composition (93.8 mole %), as indicated by the sugar composition analysis. This might be because our PLME were a crude extract that contained not only mycelium-derived components (e.g., glucans, mannans, and chitins) but also a number of constituents derived from the growth medium (e.g., glucose). In fact, the available literature shows that consumption of β-1,3:1,6-glucans is attributed to various health benefits, including immunomodulatory effects [34,35]. However, we speculate that it cannot be excluded that other types of polysaccharides (e.g., mannose- or galactose-containing polysaccharides) could also contribute to the immunostimulatory activity in PLME. In fact, the two most and well-studied immunostimulatory polysaccharides in fungi, including mushrooms, are mannan and glucan [36]. Heteropolysaccharides isolated from fungal cell walls, such as glucomannans and galactoglucomannans, which are conjugated to individual mannose, galactose, and glucose residues, have recently gained attention for improving the immune functions [36,37]. To address these possibilities, our future studies will focus on the identification of immunostimulatory polysaccharides by excluding the non-immunostimulatory portions through various fractionation and purification processes in PLME.

Collectively, our findings suggest that the consumption of PLME could enhance immune function by stimulating the systemic immune system through the spleen and intestinal immune tissues. However, our study has some limitations that need to be addressed in future studies. First, the detailed structural features of immunostimulatory polysaccharides in PLME should be characterized in terms of linkage type, molecular weight, and degree of branching. Further studies on the detailed molecular mechanism underlying immunomodulatory activity following PLME administration should be conducted.

## 4. Materials and Methods

### 4.1. Submerged Cultivation of P. linteus Mycelium

A culture stock of *P. linteus* (strain no. KCCM:60261) was obtained from the Korea Culture Center of Microorganisms (KCCM; Seoul, Korea) and activated in potato dextrose broth (PDB, Sigma-Aldrich, St. Louis, MO, USA). The seeds of *P. linteus* were inoculated onto potato dextrose agar (PDA, Sigma-Aldrich) in a Petri dish and incubated in an incubator (Hanbaek Co. Ltd., Bucheon, Korea) controlled at a temperature of 28 °C for 7 days. A piece of mycelium was aseptically transferred to a test tube filled with PDB and incubated in a shaking incubator (150 rpm; Jeio Tech Co. Ltd., Daejeon, Korea) at 28 °C for 3–5 days. Part of the cultured mycelium was collected and stored in a deep freezer (−80 °C; Ilshin Biobase, Dongducheon, Korea) until use. The remaining cultured mycelium (approximately 300 mL) was inoculated into a jar fermenter (Fermentec, Cheongju, Korea) filled with culture media [yeast extract (5 g/L; Angel Yeast, Shanghai, China), soybean flour (20 g/L; Juwon Mulsan, Cheonan, Korea), monosodium glutamate (5 g/L; Daesang, Seoul, Korea), potassium phosphate (3 g/L; ES Raw Material, Gunpo, Korea), calcium chloride (3 g/L; ES Raw Material), magnesium sulfate (1 g/L; ES Raw Material), and corn starch (40 g/L; CJ Cheiljedang, Seoul, Korea) hydrolyzed using α-amylase (Amano Enzyme, Nagoya, Japan)] (3.2 L; pH 5.0); the main submerged cultivation was performed at a temperature of 28 °C, agitation speed of 500–700 rpm and aeration of 1–1.5 vvm. Additional media (yeast extract 5 g/L, MSG 5 g/L, corn starch hydrolyzed using α-amylase 20 g/L) were added to the culture broth on days 4–5 of cultivation, and entire cultivation was allowed to continue for 7–10 days.

### 4.2. Preparation of Postbiotics Prepared from P. linteus Mycelial Submerged Culture

After submerged cultivation, the whole culture broth containing *P. linteus* mycelia was extracted at 121 °C for 60 min in an autoclave (Hanbaek Co., Ltd.). The extracts were then centrifuged at 7000× *g* rpm for 10 min, the supernatant collected and filtered through a filter paper (No. 5C, Advantec MFS, Dublin, CA, USA). The filtered solution was concentrated under reduced pressure using a vacuum concentrator (Hei-VAP Core, Heidolph Co., Bayern, Germany), lyophilized, and used in the experiments.

### 4.3. Animals and Administration Schedule

Six-week-old C3H/HeN mice (female, 17–20 g, *n* = 8) were procured from Oriental Bio Co. (Seongnam, South Korea) and bred in an experimental animal facility at Chung-Ang University, controlled at a temperature of 22 ± 1 °C, a relative humidity of 50–55%, and a light–dark cycle of 12 h. Water and common rodent feed (Dooyeol Biotech, Seoul, South Korea) were provided ad libitum. All animal experiments were approved by the Animal Experimental Ethics Committee of Chung-Ang University (IACUC NO. 2021-00032). After acclimation for a week, 40 mice were included in the study. The animals were divided into five groups (*n* = 8). The normal group (NOR) was orally administered vehicle (sterilized tap water), and the control group (CON) similarly received nutrient medium without PLME. Mice in the third to fifth groups were orally administered PLME at concentrations of 250, 500, and 1000 mg/kg/body weight (bw), respectively. The test article or vehicles were administered for 4 weeks. Throughout the experimental period, the clinical and behavioral characteristics, and mortality, if any, of all mice were monitored daily, and body weight and food consumption were recorded once a week.

### 4.4. Histomorphometric Analysis

At the end of the experiment, all mice were ethically euthanized, and the heart, kidney, liver, and spleen were collected to measure changes in the weights across the different groups. Tissue sections (4–5 μm) of the harvested organs were then processed as paraffin blocks and tissue specimens obtained according to a previously described method [38]. The tissue sections were stained with hematoxylin and eosin (H & E; Sigma-Aldrich) to evaluate histomorphometric findings. Stained tissues were observed under a microscope (Optika, Ponteranica, Italy).

### 4.5. Mitogenic Activity and Cytokine Production by Splenocytes

After euthanasia, splenocytes were isolated from the spleen tissue of all the mice according to a previously described method [8], and the cell numbers were adjusted to a density of 2.5 × 10^6^ cells/mL. The cell suspension (200 μL) was plated in a 96-well cell culture plate (Corning Costar Corp., Cambridge, MA, USA) and incubated in a humidified incubator maintained at 95% air, 5% CO_2_. Following 48 h incubation, the mitogenic activity of splenocytes was determined by the water-soluble tetrazolium salt (WST) method using EZ-Cytox (DoGenBio, Seoul, South Korea), according to the manufacturer’s instructions. In contrast, splenocyte-secreting cytokines, such as interferon-gamma (IFN-γ), tumor necrosis factor-alpha (TNF-α), and interleukin-6 (IL-6), were quantified in the supernatant of splenocytes using enzyme-linked immunosorbent assay (ELISA) kits, as per the manufacturer’s instructions. Details of the ELISA kits are provided in Appendix A.

### 4.6. Quantification of Immunoglobulin (Ig) in Serum, Small Intestinal Fluids, and Peyer’s Patch

Next, following euthanasia, serum from whole blood and luminal fluids from the small intestine (from the duodenum to jejunum) were isolated and extracted from mice according to a previously protocol described by Kim et al. [31]. Meanwhile, Peyer’s patches were collected from the wall of the small intestine of mice administered with test articles or vehicles according to a previously described method [31], and the cell numbers were adjusted to a density of 5 × 10^6^ cells/mL. The cell suspension (200 μL) was plated in a 96-well cell culture plate and incubated in a humidified incubator maintained at 95% air, 5% CO_2_ for 5 days. The levels of IgG or IgA in the serum, small intestinal fluid, and Peyer’s patch cell culture supernatant were quantified using ELISA kits, according to the manufacturer’s instructions.

### 4.7. Intestinal Immune System-Modulating Activity through Peyer’s Patch

As described above, Peyer’s patch cells were prepared, plated on a culture plate, and incubated for 5 days. Bone marrow cells were isolated from normal mice without administration of test materials, and cell numbers were adjusted to a density of 5 × 10^5^ cells/mL. To evaluate intestinal immune system-modulating activity through the Peyer’s patch, the bone marrow cell suspension (100 μL) was incubated in the presence of Peyer’s patch cell culture supernatant (100 μL) in a humidified incubator maintained with 95% air, 5% CO_2_ for 5 days [31]. The bone marrow cell proliferation activity was determined using the WST method, as per the manufacturer’s instructions. In addition, Peyer’s patch cell-secreting cytokines, such as IL-2, IL-6, and IFN-γ, were quantified in the supernatant of Peyer’s patch cells using respective ELISA kits, according to the manufacturer’s instructions. Detailed information regarding the ELISA kits is listed in Appendix A. In contrast, ileum sections (4–5 μm) of the small intestine were prepared into paraffin blocks, and the obtained tissue specimens were stained with H & E, as described above. The stained tissues were observed under a microscope to evaluate histomorphometric findings of the small intestine.

### 4.8. Analyses of General Components and Sugar Composition

The neutral sugar, uronic acid, and protein contents of PLME were determined according to the phenol-H_2_SO_4_ [39], *m*-hydroxydiphenyl [40], and Bradford [41] methods using standard references of galactose, galacturonic acid, and bovine serum albumin, respectively. The composition of monosaccharide units, rhamnose, fucose, arabinose, xylose, mannose, galactose, glucose, galacturonic acid, and glucuronic acid, was analyzed using high-performance liquid chromatography (HPLC), followed by 1-phenyl-3-methyl-5-pyrazolone (PMP) derivatization of PLME according to the method described by Hwang et al. [42]. Detailed conditions for analysis using HPLC coupled with an ultraviolet detector (HPLC-UVD) are listed in Appendix A.

### 4.9. Quantification of α- and β-Glucan Contents in PLME

The 1,3:1,6-β-glucan was determined using a commercial β-glucan assay kit for yeast and mushrooms (K-YBGL, Megazyme, Wicklow, Ireland) according to the manufacturer’s protocol. The 1,3:1,6-β-glucan content of PLME was calculated using the following formula: β-glucan (*w*/*w* %) = total glucan-α-glucan.

### 4.10. Statistical Analysis

Results are expressed as the mean ± standard deviation. Significant differences were estimated using one-way ANOVA followed by a post-hoc Tukey’s test. Statistical analysis was performed using PASW Statistics 18.0 (IBM Co., Armonk, NY, USA). Statistical significance was set at *p* < 0.05.

## 5. Conclusions

Until now, few studies and applications of postbiotics derived from mushroom mycelia under submerged cultivation have been performed. To explore the possibility of using culture broth containing *P. linteus* mycelia (PLME) as a postbiotic ingredient, we investigated the effect of PLME administration on the immune system activation in C3H/HeN mice. The present study showed that oral administration of PLME could exert immunostimulatory activity by stimulating the spleen and intestinal immune tissues in normal C3H/HeN mice. Moreover, various analytical results suggested that mannose-containing heteropolysaccharides, such as glucomannans and galactogalucomannans in addition to β-1,3:1,6-glucans, could be the bioactive candidates responsible for key role in expression of the immunostimulatory activity in PLME. To the best of our knowledge, this is the first study to show the intestinal immune system activation promoted by oral administration of PLME. In the current situation where the industrial application of mushroom mycelia is highly required, this study provides basic results and a good starting point for applying PLME as a functional ingredient to promote immune activity.

## Figures and Tables

**Figure 1 pharmaceuticals-15-01215-f001:**
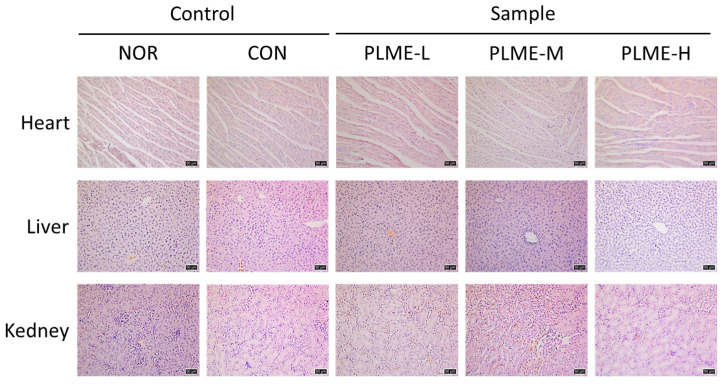
Histomorphometric analysis of major organs in mice administered with PLME for four weeks. Tissue biopsies of each organ were stained with H & E and observed under the microscope (magnification ×200). Scale bar = 50 μm. NOR, normal group administered tap water; CON, medium control group administered only culture medium (250 mg/kg/day); PLME-L, PLME-M, and PLME-H, sample groups administered postbiotics prepared from *P. linteus* mycelial submerged culture at doses of 250, 500, and 1000 mg/kg bw/day, respectively.

**Figure 2 pharmaceuticals-15-01215-f002:**
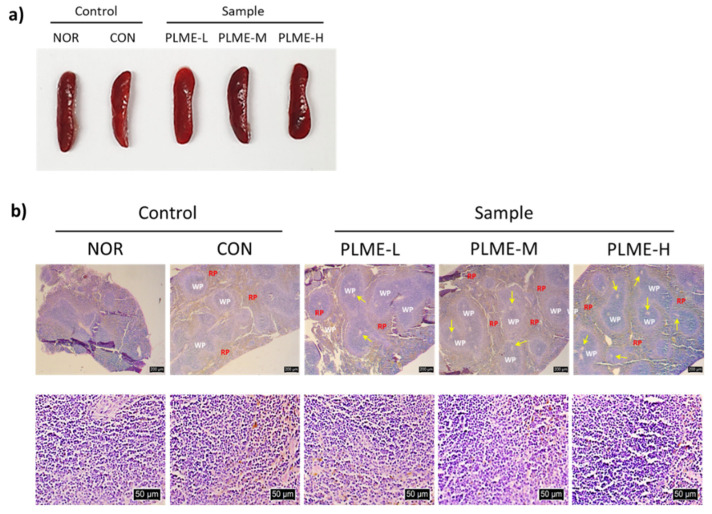
Effect of PLME administration on the spleen tissue: (**a**) Representative images of spleen tissue; (**b**) representative microscopic images (magnification ×200) of spleen biopsy stained with H & E. Scale bar = 200 μm (upper) or 50 μm (bottom). WP, white pulp; RP, red pulp; yellow arrows indicate central arteries at white pulp area of spleen tissue. NOR, normal group administered tap water; CON, medium control group administered only culture medium (250 mg/kg/day); PLME-L, PLME-M, and PLME-H, sample groups administered postbiotics prepared from *P. linteus* mycelial submerged culture at doses of 250, 500, and 1000 mg/kg bw/day, respectively.

**Figure 3 pharmaceuticals-15-01215-f003:**
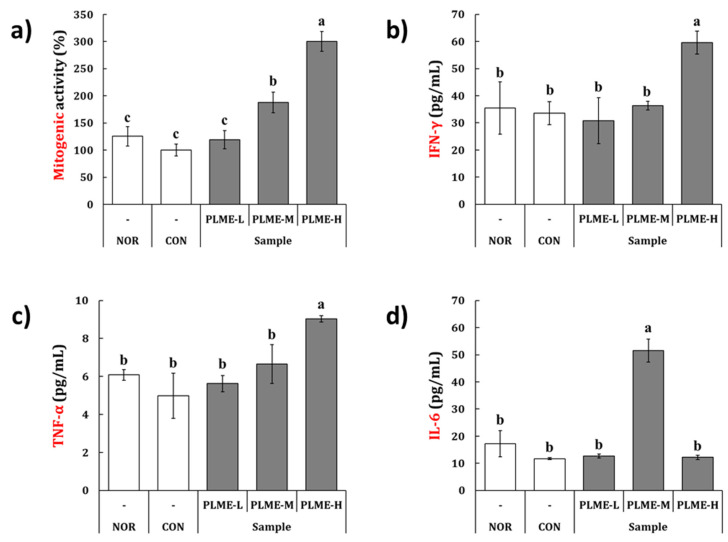
The effect of PLME administration on the (**a**) mitogenic activity and (**b**–**d**) cytokine production of the splenocytes isolated from mice. The mitogenic activity and levels of cytokines from the splenocytes were determined by WST assay and ELISA, respectively. Different letters indicate a significant difference among groups (*p* < 0.05) using Tukey’s multiple range test. NOR, normal group administered tap water; CON, medium control group administered only culture medium (250 mg/kg/day); PLME-L, PLME-M, and PLME-H, sample groups administered postbiotics prepared from *P. linteus* mycelial submerged culture at doses of 250, 500, and 1000 mg/kg bw/day, respectively.

**Figure 4 pharmaceuticals-15-01215-f004:**
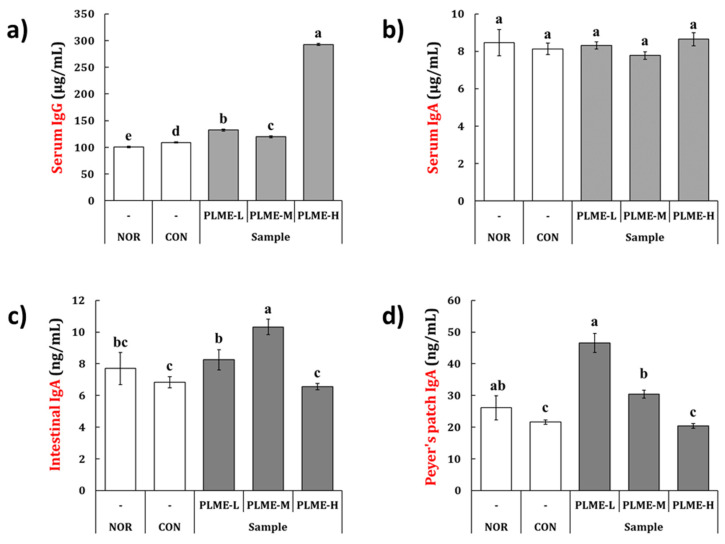
The effect of PLME administration on the level of (**a**) serum IgG, (**b**) serum IgA, (**c**) IgA in the small intestinal fluid, and (**d**) IgA in the Peyer’s patch. Each immunoglobulin was determined using ELISA. Different letters indicate a significant difference among groups (*p* < 0.05) using Tukey’s multiple range test. NOR, normal group administered tap water; CON, medium control group administered only culture medium (250 mg/kg/day); PLME-L, PLME-M, and PLME-H, sample groups administered postbiotics prepared from *P. linteus* mycelial submerged culture at doses of 250, 500, and 1000 mg/kg/day, respectively.

**Figure 5 pharmaceuticals-15-01215-f005:**
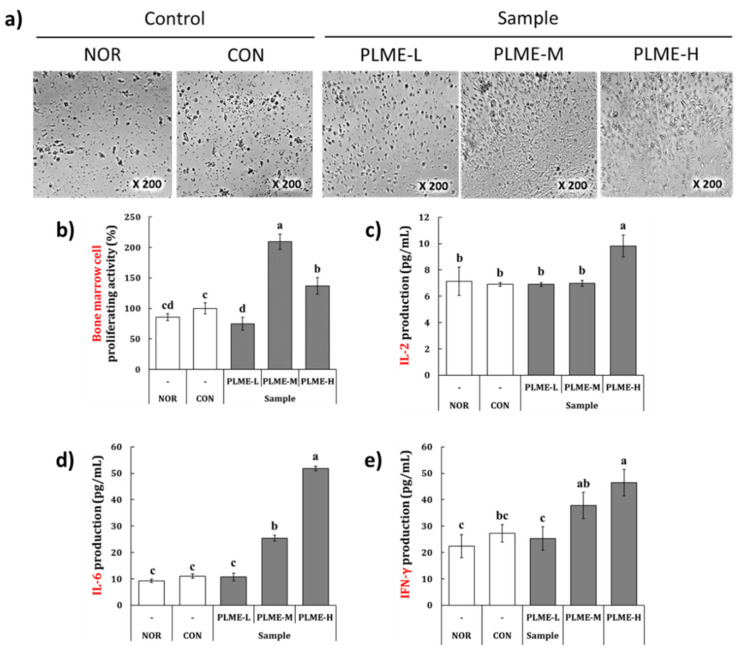
The effect of PLME administration on the intestinal immune system-modulating activity. The Peyer’s patch cells were isolated from mice administered with PLME and artificially incubated for five days, and bone marrow cell proliferation through Peyer’s patch and cytokine production from Peyer’s patch were determined by MTT assay and ELISA, respectively. (**a**) Representative microscopic images of bone marrow cells inoculated Peyer’s patch cell-cultured supernatant. (**b**) Bone marrow cell proliferating activity was determined using MTT assay. (**c**–**e**) The levels of cytokines from Peyer’s patch cells were determined using ELISA. Different letters indicate a significant difference among groups (*p* < 0.05) using Tukey’s multiple range test. NOR, normal group administered tap water; CON, medium control group administered only culture medium (250 mg/kg/day); PLME-L, PLME-M, and PLME-H, sample groups administered postbiotics prepared from *P. linteus* mycelial submerged culture at doses of 250, 500, and 1000 mg/kg bw/day, respectively.

**Figure 6 pharmaceuticals-15-01215-f006:**
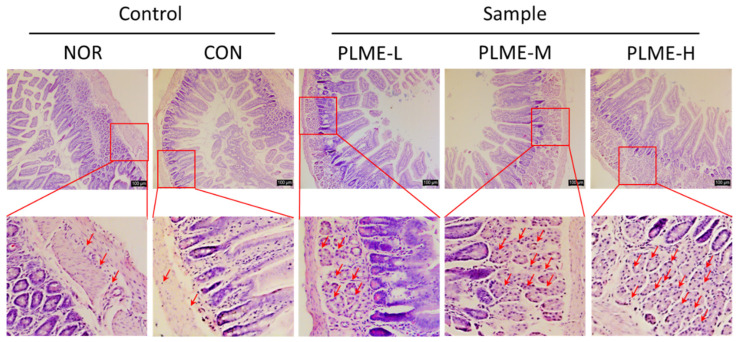
Histomorphometric analysis of small intestinal tissue in mice administered with PLME for four weeks. Tissue biopsies of small intestines were stained with H & E and observed on the microscopy (magnification ×200). Scale bar = 100 μm. Red arrows indicate recruited lymphocytes at the submucosa and muscularis externa areas. NOR, normal group administered tap water; CON, medium control group administered only culture medium (250 mg/kg/day); PLME-L, PLME-M, and PLME-H, sample groups administered postbiotics prepared from *P. linteus* mycelial submerged culture at doses of 250, 500, and 1000 mg/kg bw/day, respectively.

**Table 1 pharmaceuticals-15-01215-t001:** Weights of the major organs collected from mice administered with PLME for 4 weeks.

Group	Tissue Weight (g)		
	Heart	Kidney	Liver
NOR	0.095 ± 0.004 ^ns^	0.261 ± 0.011 ^ns^	0.901 ± 0.034 ^ns^
CON	0.105 ± 0.004	0.269 ± 0.005	0.912 ± 0.027
PLME-L	0.096 ± 0.005	0.271 ± 0.010	0.876 ± 0.037
PLME-M	0.096 ± 0.003	0.268 ± 0.004	0.881 ± 0.019
PLME-H	0.092 ± 0.002	0.264 ± 0.007	0.841 ± 0.013

Results are expressed as mean ± standard error (*n* = 8). ns, not statistically significant among all groups (*p* > 0.05). NOR, normal group administered tap water; CON, medium control group administered only culture medium (250 mg/kg/day); PLME, *P. linteus* postbiotic materials; PLME-L, PLME-M, and PLME-H, sample groups administered postbiotics prepared from *P. linteus* mycelial submerged culture at doses of 250, 500, and 1000 mg/kg body weight (bw)/day, respectively.

**Table 2 pharmaceuticals-15-01215-t002:** Chemical components, sugar, and glucan composition of PLME.

Chemical Composition	Mean ± S.D. (mg/g)
Neutral sugar	969.1 ± 18.2
Uronic acid	37.1 ± 2.2
Protein	2.2 ± 0.6
**Component sugar**	**Mean ± S.D. (mole %)**
Rhamnose	0.3 ± 0.0
Fucose	–
Arabinose	1.0 ± 0.1
Xylose	–
Mannose	1.2 ± 0.0
Galactose	2.7 ± 0.3
Glucose	93.8 ± 0.6
Galacturonic acid	1.0 ± 0.3
Glucuronic acid	–
**Glucan composition**	**Mean ± S.D. (mg/g)**
β-1,3:1,6-glucan	88.5 ± 0.9

Results were expressed as mean ± standard deviation (*n* = 3).

## Data Availability

Data is contained within the article and Appendix A.

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
