# Peer review of "Immunostimulatory Effect of Postbiotics Prepared from *Phellinus linteus* Mycelial Submerged Culture via Activation of Spleen and Peyer’s Patch in C3H/HeN Mice"

_pharmaceuticals, 2022, doi:10.3390/ph15101215_

Round 1

Reviewer 1 Report

Research paper entitled "Immunostimulatory Effect of Postbiotics Prepared from Phellinus linteus Mycelial Submerged Culture via Activation of Spleen and Peyer’s patch in C3H/HeN Mice" addresses the hot topic of postbiotics and a novel and poorly explored source of postbiotics, which are mushrooms. The introduction provides sufficient info, however, it can be improved. The experimental design is good and the methods are well and sufficiently explained. Results are presented in a clear way and support the discussion. The greatest value of this work is that it deals with understudied material and provides valuable info for the next stage of studies (clinical ones) as well as safety data for the area in which they are missing. 

Minor corrections should be done: 

Line 42: erase "belonging to the fungal kingdom" 

Line 48: sclerotia instead of sclera bodies

Line 49: health promoting purposes instead of health functional purposes

Line 51: various industries? Not so sure. Can you provide examples?

Lines 344-358: try to avoid repetition of "immune system". It appears in every sentence.

It would be a pleasure to read your next work based on the promises stated at the end of the paper. Good luck!

Reviewer 2 Report

Medicinal mushrooms are higher fungi with nutraceutical properties and are low in calories and fat. The research manuscript entitles "Immunostimulatory Effect of Postbiotics Prepared from Phellinus linteus Mycelial Submerged Culture via Activation of Spleen and Peyer’s patch in C3H/HeN Mice" by Kim, et al., evaluated the effect on the immune system of postbiotic materials (PLME) from Phellinus linteus mycelial submerged cultureusing normal C3H/HeN mice. The results showed an increase in immune cells in the spleen and small intestinal tissues of mice administered PLME, supporting the rationale for its immune system activation. The manuscript has been written regular with a good discussion. In general, the research concept is timely and of interest to the readers of natural active compound fields particularly from medicinal mushrooms. There are some revisions for better understanding as below:

- L199, what is the total glucan measurement method? Please provide it.

- Conclusions section:  The authors should provide more information for Conclusions section. This section is so simple and short.
